# Dietary Intakes of Zinc, Copper, Magnesium, Calcium, Phosphorus, and Sodium by the General Adult Population Aged 20–50 Years in Shiraz, Iran: A Total Diet Study Approach

**DOI:** 10.3390/nu12113370

**Published:** 2020-11-01

**Authors:** Elham Babaali, Samane Rahmdel, Enayat Berizi, Masoumeh Akhlaghi, Friedrich Götz, Seyed Mohammad Mazloomi

**Affiliations:** 1Department of Food Hygiene and Quality Control, School of Nutrition and Food Sciences, Shiraz University of Medical Sciences, 71645-111 Shiraz, Iran; elhambabaali68@gmail.com (E.B.); Rahmdel.samane@gmail.com (S.R.); Enayat.berizi@gmail.com (E.B.); 2Department of Community Nutrition, School of Nutrition and Food Sciences, Shiraz University of Medical Sciences, 71645-111 Shiraz, Iran; msm.akhlaghi@gmail.com; 3Nutrition Research Center, School of Nutrition and Food Sciences, Shiraz University of Medical Sciences, 71645-111 Shiraz, Iran; 4Microbial Genetics, Interfaculty Institute of Microbiology and Infection Medicine Tübingen (IMIT), University of Tübingen, D-72076 Tübingen, Germany; friedrich.goetz@uni-tuebingen.de

**Keywords:** daily dietary intake, essential elements, risk assessment, total diet study, Iran

## Abstract

In the present total diet study, the dietary intake of zinc (Zn), copper (Cu), magnesium (Mg), calcium (Ca), phosphorus (P), and sodium (Na) by healthy adults in Shiraz, Iran, was estimated from the foods as consumed. A total of 580 individual food items were collected, prepared, and pooled into 129 composite samples. The metal concentration was then evaluated using inductively coupled plasma–optical emission spectrometry. The mean intakes of Zn (12.92 mg/d), Cu (3.80 mg/d), and Mg (412.68 mg/d) exceeded the estimated average requirements (EARs), but they were well below the upper limits. A high prevalence of inadequate intake was observed for Ca (91.6%) and P (89.7%), which was mainly due to nutritionally imbalanced diets. Sodium intake for average and high consumers (97.5th percentile) was 123.6% and 237.8% of the tolerable upper intake level of 2300 mg/d, respectively, with 70% of the participants having intakes higher than this threshold value. Nutrition education, nutritional rehabilitation, Ca supplementation, food fortification, mandatory reduction of salt content in processed foods, and discretionary salt use (in home cooking or at the table) are among the possible strategies that can be adopted to combat the health problems.

## 1. Introduction

Diet, as a source of nutrients and contaminants, is a crucial contributor to human health and disease [1]. Although nutrient deficiency is one of the major health concerns in both developed and developing countries, the possibility of adverse health effects arising from excessive intake should not be overlooked [2]. Minerals are dietary components required for normal growth, development, and body homeostasis. Among the most important minerals, zinc (Zn) is an essential element for human metabolism that functions as an enzyme cofactor, contributes to protein structure, and regulates gene expression [3]. Its deficiency is associated with immune dysfunction, inflammation, oxidative stress, increased risk of infection, cardiovascular diseases (CVD), and osteoporosis. Copper (Cu) is an integral part of antioxidant enzymes and its deficiency has been implicated as a risk factor for impaired immune function and ischemic heart disease [4]. Magnesium (Mg) is another physiologically important element that plays a key role in muscle contraction, gland secretion, and nerve transmission. It also has protective effects against CVD by enhancing endothelium-dependent vasodilation, improving lipid metabolism and profile, reducing systemic inflammation, and inhibiting platelet aggregation. Calcium (Ca) has been reported to exert cancer-preventive effects and its contribution to bone health is well established. Meanwhile, a high circulating level of Ca has been proposed as a risk factor for CVD [3]. As the second most abundant element in the human body, phosphorus (P) is an essential component of bone mineral, cell membrane, nucleic acids, and cellular energetic metabolites [5]. Excessive dietary intake of P can affect the absorption and metabolism of Ca and Mg, contributing to the development of osteoporosis [6]. Sodium (Na) is a key electrolyte for the regulation of blood pressure, blood volume, and blood osmolality. High dietary intake of Na, however, is related to a higher risk of hypertension, CVD, and kidney disease [3].

Dietary exposure assessments may give clues to assess the health risks associated with mineral deficiency and toxicity [7]. To date, different approaches have been proposed to evaluate the dietary intake of nutrients. Although there is not, and probably never will be, a method that can estimate dietary intake without error [8], some assessment methods have some advantages over the others in terms of reliability. Using food composition tables (FCTs), together with individual surveys, such as food frequency questionnaire (FFQ), 24-h dietary recall, dietary history, and the dietary record, is still the most prevalent method applied in nutrition studies. The FCT-based estimates, however, are subject to large uncertainties. These uncertainties become more significant when foreign FCTs are employed that are not representative of local foods and culinary culture [9].

Due to the lack of comprehensive data on the nutrient content of Iranian foods, the majority of national nutrition surveys in Iran are generally conducted using foreign FCTs, which, in turn, can adversely affect the accuracy of dietary intake measurements. In our previous studies on the dietary intakes of Cu, P, and iron by adults living in Shiraz, Iran, we observed a discrepancy between the data obtained from instrumental analysis and FCT-based dietary assessment tools [7,9,10]. A useful public health tool recommended by the World Health Organization to assess dietary exposure to chemical substances across the entire diet is the total diet study (TDS) [11,12], which, despite providing more reliable exposure estimates [2], has not been employed heretofore in national nutrition surveys conducted in Iran. In TDS experiments, foods representing the average diet of the study population are purchased at the retail level, prepared as consumed, and combined into food composites prior to the analysis of nutrients of interest. The analysis of the foods as consumed yields more refined estimates of dietary exposure [2,13,14,15]. Of the methods available for the elemental analysis of food matrices, inductively coupled plasma (ICP) techniques are the most commonly used and recommended due to their advantages, including high sensitivity, high selectivity, and multi-element detection capability [16,17,18]. The potential of ICP-based techniques in the simultaneous multi-element analysis of a high number of food samples within a short period of time allows the timely adoption of measures to address nutritional problems. The lower detection limits of these techniques allow the detection of very low levels of trace elements in food products and identification of small changes in mineral content of food following different methods of preparation [19].

The present study aimed to provide the first TDS results on the dietary exposure of the general adult population in Shiraz, Iran, to Zn, Cu, Mg, Ca, P, and Na using inductively coupled plasma–optical emission spectrometry (ICP-OES) as a detection technique.

## 2. Materials and Methods

### 2.1. Food Consumption Data

The consumption data were derived from the nutrition survey carried out between November 2013 and March 2014 in the city of Shiraz (with 1,500,000 inhabitants and 416,000 households), Fars province, Iran [20]. In brief, dietary intake was measured in a representative sample of adult residents of Shiraz consisting of 438 healthy individuals (199 males and 239 females), aged 20–50 years, using a 160-item FFQ, the validity and reliability of which was established previously [21,22]. The interview respondents were selected from households that had been stratified into nine municipal districts using a stratified, multistage, random sampling [20]. The ratio of males to females in the study sample roughly corresponded to the gender distribution in the general population of Shiraz (total non-elderly adult population ~790,000).

Both food items with the average consumption rate of more than 1 g/d and commodities known or suspected to contribute significantly to the intake of the elements tested were selected as representative food items [23]. In total, 116 core foods were selected and categorized into 20 groups (bread; cereals and cereal products; legumes; red meat and meat products; poultry; fish; eggs; milk and milk products; raw vegetables; cooked vegetables; potatoes; fruits; fruit juices and soft beverages; nuts and dried fruits; cookies, cakes, and pastry; snacks; fats and oils; honey and sugar; condiments; and drinking water) for further analysis (Appendix A).

### 2.2. Sample Collection and Preparation

A total of 580 food samples (five items per each food product) were randomly purchased from retail outlets, markets, and bakeries located in five geographical districts of the city from October 2015 to June 2016. The bulk foods were sampled in portions of 250 g. The packaged foods were collected from different brands with similar production dates. The water samples related to each district were collected from domestic drinking water plumbing systems.

On each sampling day, the food items were individually prepared according to the local food consumption habits but without adding any salt or additives. Where required, a pooled water sample from the five districts was used for food preparation. The preparation of samples was conducted using stainless steel instruments to avoid any possible cross-contamination. Afterward, five food items of the same type with the same preparation were pooled at equal proportions (100 g) and homogenized in a commercial kitchen blender equipped with an Eastman Tritan copolyester jug and stainless-steel blades (JTC Electronics Corp., OmniBlend I series, Model TM-767, Zhongshan, Guangdong, China). A total of 129 composite samples were prepared and stored at −20 °C until further analysis.

### 2.3. Chemical Analysis

#### 2.3.1. Materials

All chemicals and reagents were of either analytical or ultrapure grade from Merck Co. (Darmstadt, Germany).

#### 2.3.2. Sample Digestion

A portion of fresh sample (500 mg) was digested in 60 mL Teflon vessels, with 3.5 mL of 65% nitric acid and 1.5 mL of 30% hydrogen peroxide, using a closed-vessel microwave digestion system (Sepehri, Shiraz, Iran) and made up to 25 mL with deionized water. The blanks were prepared in the same way as the main samples containing the reagents, but with no homogenate [10].

#### 2.3.3. ICP-OES Analysis

The metal concentration was measured by ICP-OES (Spectro Arcos, SPECTRO Analytical Instruments GmbH, Kleve, Germany) under the experimental conditions summarized in Table 1.

#### 2.3.4. Quality Control

Analytical quality was verified using standard calibration solutions, blanks, blind replicates, and spiked samples (at 50% of the original concentration) at every measurement. The linearity of the calibration curves estimated by the square correlation coefficient (r^2^) was greater than 0.999. The mean blank value was subtracted from the sample concentrations. For all the analytes, the relative standard deviation (RSD) of the replicates was below 20%. The average recovery rates obtained for Zn, Cu, Mg, Ca, P, and Na were 91.72, 87.08, 83.60, 84.97, 83.06, and 94.34%, respectively. The limits of quantification (LOQ, mg/kg) were 0.9 for Zn, 1.0 for Cu, 4.3 for Mg, 2.3 for Ca, 3.2 for P, and 171 for Na. The limits of detection (LOD, mg/kg) obtained for Zn, Cu, Mg, Ca, P, and Na were 0.27, 0.3, 1.29, 0.69, 0.96, and 51.3, respectively.

### 2.4. Exposure Assessment and Risk Characterization

For each composite sample, the metal contents were expressed in milligrams per kilogram of fresh weight and the results were then aggregated at the food group level. The dietary intake levels of the elements from each food group were estimated on the basis of the concentrations of the selected elements and food consumption data. The contribution of food groups to the total element intake was then calculated.

The 2.5th (PCT2.5) and 97.5th (PCT97.5) percentiles of the estimated daily intake levels were used to represent low and high consumers, respectively. The mean probability of adequacy was calculated by comparing the intake levels with the estimated average requirement (EAR) or adequate intake (AI) values proposed by the Institute of Medicine [24,25,26,27]. The intake values were also compared with the tolerable upper intake levels (ULs) to evaluate the possible health risks (Table 2).

### 2.5. Statistical Analysis

Data were analyzed using IBM SPSS 21.0 statistical software (IBM Corp., Armonk, NY, USA). Comparisons were performed using one-way analysis of variance followed by the Duncan test. The level of significance was set at 0.05.

## 3. Results

### 3.1. Mineral Concentration

The levels of the investigated elements in all the samples were above the LOD values. The concentrations of elements for each food group are summarized in Table 3. The highest level of Zn was found in “red meat and meat products”, with a mean value of 31.68 mg/kg fresh weight, but significantly lower contents were detected in other food groups. Among all food groups, the “nuts and dried fruits” group was the richest source of Cu (6.90 mg/kg) and Mg (1250.45 mg/kg). The “milk and milk products” contained the highest Ca concentration (543.43 mg/kg), followed by the “nuts and dried fruits” (405.68 mg/kg). As expected, the main dietary sources of P were protein-rich foods, including “poultry”, “red meat and meat products”, “fish”, and “nuts and dried fruits”. Significant variations were observed between different food groups regarding the Na levels, with the highest values found in the “condiments”, “snacks”, “red meat and meat products”, “fish”, “poultry”, “bread”, and “cereal and cereal products”.

### 3.2. Dietary Exposure Estimates

The average dietary exposures to six elements from different food groups are presented in Table 4. Figure 1 shows the contribution of 20 food groups to the daily intake of these elements. The main contributors to dietary exposure to Zn were “drinking water” and “red meat and meat products” (33.6% and 14.0%, respectively). The contribution of other groups was less than 10%. The food groups contributing most to the dietary intake of Cu (3.80 mg/d) were “drinking water” (26.5%) and “fruits” (21.0%), followed by “raw vegetables” (10.0%). “Raw vegetables”, along with the “milk and milk products” and “fruits” groups, made the greatest contribution to dietary Mg intake of 412.68 mg/d (19.6, 14.9, and 14.8%, respectively). “Drinking water” and “milk and milk products” accounted for 36.2 and 25.6% of the daily dietary exposure to Ca (532.61 mg), respectively, followed by “raw vegetables” and “fruits” (10.9 and 8.9%, respectively). The other groups contributed less than 4% to the mean exposure. “Bread” and “milk and milk products” showed the highest contribution to P intake (15.9 and 15.4%, respectively). “Condiments”, “cereals and cereal products”, “bread”, “drinking water”, and “milk and milk products” contributed to 83.8% of the dietary exposure to Na (43.6, 13.9, 9.0, 8.8, and 8.5%, respectively). “Snacks”, “fats and oils”, and “honey and sugar” were found to make the least contribution to the dietary intake of the majority of tested elements.

### 3.3. Risk Characterization

The mean and the 2.5–97.5th percentile range of total dietary intake of each element, along with the related health risks, are summarized in Table 5. The daily dietary intakes of Zn, Cu, and Mg were higher than the EAR values, but still far lower than the tolerable upper intake levels. However, the total sodium intake exceeded the UL of 2300 mg/d in 69.6% of the participants. The mean intakes of Ca and P were less than 70% of the EAR values, with a high proportion of the study population (more than 89%) at risk of inadequate intake. The estimated 2.5th percentile exposures to Ca and P were 30.6 and 35.4% of their respective EAR values, respectively. Na was the only element with the mean and 97.5th percentile intake higher than the UL.

## 4. Discussion

Although different methods have been proposed to evaluate the nutritional status of a certain population, a total diet study is by far the most accurate method [11]. In Iran, as in many other less-developed countries, the information available on the metal content of the foodstuff is scarce [7]. The vast majority of data on dietary intakes of elements have been, therefore, generated using FCT-based nutrition surveys, which are not reliable enough to estimate the risks of deficiency and toxicity. Herein, the dietary exposure of six essential elements from foods commonly consumed by the general adult population in Shiraz, Iran, was estimated using a TDS approach.

### 4.1. Zinc

Although foods are assumed to be the main sources of essential mineral nutrients, due to the high consumption, drinking water can also significantly contribute to the intake of minerals in some communities [28]. In the present study, the “red meat and meat products” food group, despite having the highest Zn concentration, was the second contributor to the dietary intake of Zn only after drinking water. The contribution of different food groups to nutrient intake is highly dependent on the dietary habits of the study population. While meat and its products have been identified as the major source of dietary exposure to Zn in many countries [2,14,15,29,30,31,32,33,34,35], in some other regions, other food groups, such as dried and smoked seafood, grains, and cereals, contribute most to the dietary intake of Zn [36,37]. The Zn content in animal foods has been suggested not to be dependent on food origin or affected by the cooking process, which can mainly be attributed to the lower proportion of water-soluble Zn in these products compared to plant foods [7,38]. It, therefore, was not unexpected that the concentrations of Zn in foods of animal origin in this study were in the same range reported for other regions of the world [2,11,14,15,29,30,32,39,40,41].

The mean daily intake of Zn by the participants in this study was well comparable to the intake value obtained in our previous study on the same population using the analysis of duplicate portions of all food items, excluding snacks, drinking water, and beverages (9.39 mg/d). In spite of exceeding the EAR values, the average dietary intake of Zn was unlikely to be of toxicological concern in the Shiraz population. Similar TDS-based intake levels have been reported for Korean (10.16 mg/d) [37], Nigerian (8.7 mg/d) [41], Lebanese (10.97 mg/d) [2], British (9.9 mg/d) [15], Italian (10.6–12.0 mg/d) [31,33,35], and French (8.66 mg/d) [14] adults, which are slightly higher than the estimates reported for their Cameroonian (6.54 mg/d) [36] and Chilean (7.6 mg/d in males and 6.4 mg/d in females) [32] counterparts. However, much higher values have been reported for only a few countries, including Saudi Arabia (17.7 mg/d) [34] and Spain (21.8 mg/d) [42].

### 4.2. Copper

Drinking water has been shown to have a considerable contribution to the dietary intake of Cu in Fars province [28], which is in good agreement with the results of this study. Among the study food groups, fresh fruit and vegetables contributed to 31% of the dietary exposure to Cu. A similar contribution of fruit and vegetables to Cu intake was reported in Italy (35–39%) [31,33,35]. Substantial but lower contribution levels, however, were found in other regions [2,14,15,32]. The Cu content in plant and animal foods varies depending on the environmental conditions and dietary habits of meal preparation [7], which provides an explanation for some differences observed in the Cu concentrations between the samples collected from different regions of the world [2,11,14,15,30,32,34,41,42].

The daily intake of Cu in the Shiraz population (3.8 mg/d) was much greater than that estimated previously for the same population using duplicate portion sampling (DPS) (1.19 mg/d) [7]. This discrepancy may be due to excluding snacks, drinking water, and beverages in the DPS-based estimation. Moreover, the present DPS study only targeted the representative diet of the middle-income group; therefore, the results may not provide a precise estimation of the population intake. The mean intake estimates range from 0.98 mg/d in France to 2.7 mg/d in Nigeria [2,13,14,15,31,33,35,41,42]. Although in this study, the mean intake of Cu was more than five times higher than the EAR value, only 0.5% of the study population had intakes higher than the UL value, which cannot be a health concern.

### 4.3. Magnesium

The available data on Mg concentrations in food products from different countries around the world show some discrepancies [11,14,30,43,44], which can be well explained by the fact that the content of this element is highly influenced by the food origin and processing [45]. The highest Mg content was found in nuts and oilseeds, which have also been reported to be rich sources of Mg in other total diet studies [11,14,43]. However, due to the very low average daily intake of these food products, they contributed to only around 4% of the dietary exposure to Mg, while “raw vegetables” showed the highest contribution. The main contributor to the dietary intake of Mg may differ by region depending on the habitual diet of the study population. Nevertheless, a review of the available literature revealed that vegetables were among the food groups contributing most to the dietary intake of Mg [14,31,35,36,44].

The mean daily intake of Mg among male and female adults was slightly higher than the EAR values (130 and 142% of the EAR, respectively). The percentage of the population with an intake less than the EAR values was estimated to be 22.4%. Lower intake of Mg has been reported in some countries/regions, including Yaoundé, Cameroon (294 mg/d) [36], France (224 mg/d) [14], Italy (262 mg/d) [35], and Andalusia, Spain (366 mg/d) [44]. However, there is no evidence suggesting that high intake of Mg naturally occurring in foods can have toxic effects. It must be noted that the UL applies only to the supplementary Mg intake [46].

### 4.4. Calcium

The highest Ca concentration was found in “milk and milk products”, among which the UF-feta cheese was the richest in calcium (1520.68 mg/kg), which may be mainly due to the concentrating effect of milk coagulation and dehydration during cheese-making [43,44]. A comparison of the results from total diet studies conducted worldwide showed large discrepancies in the Ca content of the same food items [11,14,30,43,44]. The concentration of this macro-mineral in plant foods depends on the levels of Ca in the soil and Ca assimilation rate. Moreover, technological and cooking processes can affect the Ca content in these products [44,47]. There are also some factors that can affect the content of Ca in cow milk; these factors include breed, within-breed genetic diversity, health condition, the environment, and management practices [48].

The average Ca intake in the study population was comparable to the estimates reported by Lee et al. [37] for Koreans (460.6 mg/d). Higher intake of Ca has been reported in total diet studies conducted in Cameroon (760 mg/d) [36], France (721 mg/d) [14], Italy (738 mg/d) [35], Spain (1266.6 mg/d) [44], and the UK (747 mg/d) [49]. In all these studies, milk and dairy products were found to be the main contributors (from 29.6% in Korea to 59% in Italy) to the dietary intake of Ca. The only exception was the study carried out in Cameroon, where the fish group provided 65% of the daily Ca intake while the dairy products contributed only to 3% of the total intake. This result can be explained by the very low consumption of dairy foods by Cameroonian adults (10 g/d) [36]. In the current study, drinking water contributed most to the intake of Ca. A much lower contribution of water to Ca intake has been reported in Italian and Spanish adults (9 and 10%, respectively) [35,44]. Nearly 92% of the participants were at risk of inadequate intake (<EAR), which can be attributed to the low per capita consumption of milk and dairy products in the study population. The average annual consumption of milk per person in Iran is 50 kg, which is by far lower than the values reported in western countries [50]. A national comprehensive study on the pattern of household food consumption and nutritional status in Iran showed a high prevalence of inadequate Ca intake in all Iranians [51]. Given the health implications of Ca deficiency, such a low intake level is highly concerning and deserves much more attention by both researchers and health policy-makers.

### 4.5. Phosphorus

Foods of animal origin, including poultry, meat, and fish, as well as protein-rich plant food products, such as nuts and legumes, are the main sources of dietary P [52]. In this study, although these food groups had the highest P concentrations, the extent to which they contributed to the daily P intake of the study population was lower than that of bread and dairy products. Similar findings were reported in an Italian total diet study, in which the milk and dairy food group contributed 27% and cereals and cereal products 22% of the total dietary intake of P [35]. The P content in the food items selected for the present study was significantly different from the values reported in a total diet study conducted in the US [30]. These discrepancies could be attributed to differences in agricultural practices, such as manuring and application of phosphorus fertilizer, food processing, and culinary activities. One example of such differences is the use of trisodium phosphate for decontamination of poultry in the US [53] while the current Iranian legislation does not allow chemical decontamination of foods of animal origin [9]. Furthermore, there is some evidence suggesting that a considerable amount of this element is lost during cooking of peas, fish, and meat [9].

The daily P intake in the participants ranged from 180.03 to 1093.15 mg/d. P was the mineral with the second-highest prevalence of inadequate intake (89.7%). The available data on dietary exposure to P are generally from individual surveys combined with FCTs, but rarely from DPS and TDS studies, and all have reported daily P intakes that exceed the EAR of 580 mg/d [9,35]. The quantitative differences in consumption profiles and wide variation in dietary assessment methods may provide an explanation for this apparent discrepancy between the results. In our previous attempt to estimate the dietary intake of P in adults living in Shiraz, duplicate diet samples of 21 different complete meals (excluding snacks, drinking water, and beverages) were collected from a hospital kitchen for seven consecutive days. The mean intake from this representative diet of the middle-income population was estimated to be 1200.3 ± 236.0 mg/d [9]. Compared with the results of the present study, higher daily consumption of P-rich foods was found in our previous seven-day duplicate portion study as follows: 206.6 vs. 80.4 g/d of meat and meat alternates (including red meat, poultry, fish, soy, and egg), 340.6 vs. 274.4 g/d of milk and dairy products, and 744.0 vs. 390 g/d of bread and cereals. The main drawback of our DPS study was that we did not take into account the wastage of meal samples. Accordingly, and based on the DPS study protocol, we could only sample the meals representative of the reference person’s food intake, but not of the whole population.

### 4.6. Sodium

In agreement with other studies [11,43,54], processed foods had higher Na content than their unprocessed counterparts in the present study. For example, the Na content of pickled cucumber and tomato paste were 76 and 162 times higher than that of fresh cucumber and tomato (2447.51 and 2421.86 mg/kg vs. 32.16 and 14.94 mg/kg, respectively). Na inherent in foods and that added in food processing contributed 56.4% (1.60 mg/d) and salt added at the table or during home cooking 43.6% of the dietary exposure to Na. The contribution of salt to total Na consumption in adults was estimated to be 52% in Italy [35] and up to 20% in New Zealand [54]. In western diets, the major source of Na intake is processed foods rather than table/cooking salt or unprocessed products [43,54].

The TDS-based estimates of Na intake worldwide show a narrow range of variation from 2100 mg/d in Cameroon (excluding table and cooking salt), 2150–3603 mg/d in New Zealand, 2300 mg/d in France, 2843 mg/d in this study, to 3812 mg/d in Italy (considering the discretionary salt) [14,35,36,54]. In this study, only 120 participants (27.4%) had intakes between the AI and UL, while the US UL of 2300 mg/d was exceeded in around 70% of the cases. For a high consumer (PCT97.5), Na exposure was more than twice the UL value. Our results showed that Na from processed and unprocessed foods represented more than 100% of the required level. Therefore, the discretionary use of salt in the household seems to have a relatively large contribution to the overall intake of Na and can be a key target for Na reduction.

## 5. Conclusions

A TDS study was conducted to estimate the dietary exposure of the general adult population of Shiraz to six essential trace elements and assess the risks associated with nutrient deficiency and excessive intake. The results can be to some extent extrapolated to other urban areas. The rural population, however, has a different habitual diet, which, in turn, may lead to different nutrient intakes. With respect to Zn, Cu, and Mg, the prevalence of inadequate or excessive intakes was too low to be of major concern. Meanwhile, a very high proportion of the participants (more than 89%) were at risk of inadequate intake of Ca or P, mainly due to the low consumption of animal foods, including milk and dairy products. Although inadequate Ca intake is a national health problem in Iran, there is a lack of efficient strategies to address Ca deficiency. National programs integrating nutrition education and rehabilitation, Ca supplementation, and food fortification need a strong governmental impetus to put them into practice. The mean and high (97.5th percentile) intakes of Na exceeded the upper intake limit by factors of 1.2 and 2.4, respectively. Almost 70% of the participants had intakes higher than the UL value, a finding which must be given special heed. Daily salt intake in the Iranian population substantially exceeds the current recommendations. Thus, national prevention efforts should be directed at reducing the use of salt in commercial food processing or domestic cooking and at the table. Improving food labeling is another health strategy that may help consumers to make a healthy food choice with lower Na content.

This is the first-ever study on dietary intakes of elements using a TDS approach in Iran. The study population, nevertheless, was limited to one metropolitan area. Hence, the results may not be generalizable to other regions of the country, given regional variations in dietary patterns and food habits. Moreover, the exclusion of populations other than adults limits the generalizability of findings to the whole population. This highlights the need for additional total diet studies on the adequacy of dietary intake in population groups other than healthy adults in other regions not studied yet. To ensure the safety and nutritional adequacy of food supplies, it is necessary to examine and evaluate the dietary exposure to a wide range of food chemicals, however.

## Figures and Tables

**Figure 1 nutrients-12-03370-f001:**
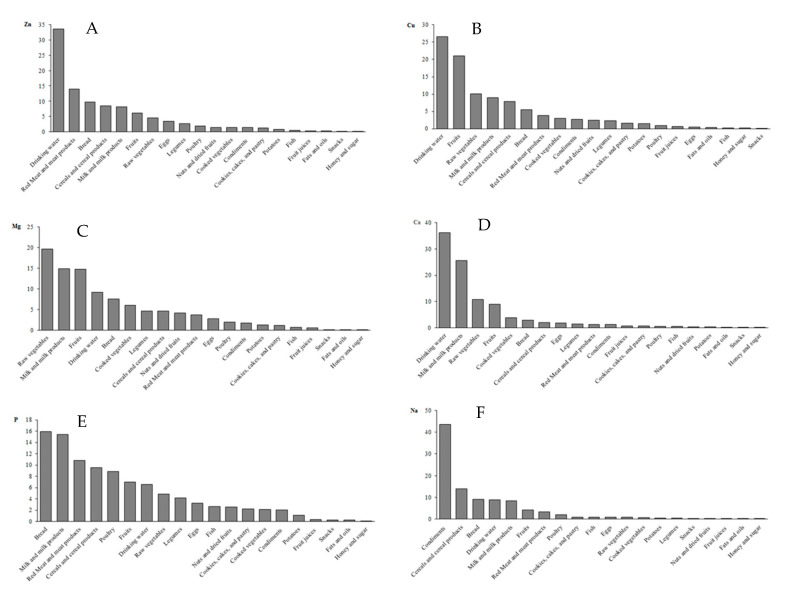
The mean contribution (%) of food groups to dietary intakes of (**A**). zinc (Zn), (**B**). copper (Cu), (**C**). magnesium (Mg), (**D**). calcium (Ca), (**E**). phosphorus (P), and (**F**). sodium (Na) in adult population in Shiraz, Iran.

**Table 1 nutrients-12-03370-t001:** Inductively coupled plasma–optical emission spectrometry (ICP-OES) instrumental and operational parameters for metal determination.

Parameter	Value/Type
Radio-frequency generator (W)	1400
Plasma torch	Auxiliary
Nebulizer gas	Argon
Plasma gas flow rate (L/min)	14.5
Auxiliary gas flow rate (L/min)	0.9
Nebulizer gas flow rate (L/min)	0.85
Sample uptake time (S)	240 total
Rinse time of (S)	60
Initial stabilization time (S)	Preflush:60
Measurement replicate	3
Element (λ/nm)	As below
Frequency of RF generator (MHz)	Resonance frequency: 27.12 MHz
Type of detector Solid state	Charge coupled device (CCD)
Type of spray chamber Cyclonic	Cross flow

**Table 2 nutrients-12-03370-t002:** Nutritional reference values (mg/d) for zinc (Zn), copper (Cu), magnesium (Mg), calcium (Ca), phosphorus (P), and sodium (Na).

Guidance Values	Elements					
	Zn	Cu	Mg	Ca	P	Na
EAR ^a^	Male: 9.4 Female: 6.8	0.7	Male: 350 * Female: 265 *	800	580	1500 **
UL ^a^	40	10	NA	2500	4000	2300

^a^ Estimated Average Requirement (EAR) values and Tolerable Upper Intake Levels (ULs) for adults (19–50 years) established by the Institute of Medicine [24,25,26,27]. * EAR values for adults aged 31–50 years (the mean age of male and female participants was 34 and 36 years, respectively). ** Adequate Intake (lack of sufficient scientific evidence to establish an EAR). NA, not applicable as no UL is set for dietary Mg intake.

**Table 3 nutrients-12-03370-t003:** The mean, minimum, and maximum concentrations (mg/kg fresh weight) of zinc (Zn), copper (Cu), magnesium (Mg), calcium (Ca), phosphorus (P), and sodium (Na) in foods “as consumed” at food group level in Shiraz, Iran Total Diet Study.

Food Group	Elements					
	Zn		Cu		Mg	
	Mean ± SE	Min–Max	Mean ± SE	Min–Max	Mean ± SE	Min–Max
Bread	9.64 ± 0.82AC	7.58–11.23	1.67 ± 0.17AB	1.30–2.01	248.27 ± 24.49A	177.78–286.83
Cereals and cereal products	3.85 ± 1.15AC	1.86–8.13	0.70 ± 0.36AB	0.05–1.96	69.39 ± 7.10A	45.63–90.12
Legumes	11.22 ± 1.63AC	7.09–15.43	3.02 ± 0.60AB	1.80–4.76	523.57 ± 140.54A	290.13–1074.25
Red meat and meat products	31.68 ± 8.42B	5.48–66.38	4.46 ± 2.78BC	0.76–21.11	278.68 ± 53.83A	91.63–490.09
Poultry	10.17 ± 2.30AC	7.86–12.48	1.37 ± 0.25AB	1.12–1.62	356.34 ± 97.68A	258.67–454.02
Fish	5.65 ± 0.28AC	5.37–5.94	0.92 ± 0.05AB	0.87–0.98	234.96 ± 85.46A	149.50–320.42
Eggs	11.33 ± 0.00AC	11.33–11.33	0.43 ± 0.00A	0.43–0.43	297.01 ± 0.00A	297.01–297.01
Milk and milk products	4.42 ± 0.75AC	2.09–10.67	1.24 ± 0.16AB	0.68–2.41	222.02 ± 51.83A	52.78–491.24
Raw vegetables	2.13 ± 0.40A	0.48–4.58	1.48 ± 0.15AB	0.75–2.06	236.00 ± 56.75A	57.28–485.98
Cooked vegetables	3.14 ± 0.67AC	0.33–5.84	1.78 ± 0.34AB	0.66–4.21	303.61 ± 80.22A	74.28–764.37
Potatoes	3.33 ± 0.05AC	3.28–3.38	1.96 ± 0.15AB	1.81–2.11	192.35 ± 29.78A	162.58–222.13
Fruits	1.28 ± 0.19A	0.03–3.23	1.61 ± 0.11AB	1.01–3.61	114.87 ± 19.09A	42.38–396.09
Fruit juices and soft beverages	1.26 ± 0.41A	0.25–2.71	0.76 ± 0.10AB	0.50–1.14	192.24 ± 125.69A	8.03–686.66
Nuts and dried fruits	14.06 ± 7.13C	1.18–38.18	6.90 ± 2.21C	1.56–12.91	1250.45 ± 810.68B	173.23–4465.23
Cookies, cakes, and pastry	5.38 ± 0.33AC	4.02–6.36	1.80 ± 0.39AB	0.55–3.23	133.51 ± 21.41A	82.68–223.70
Snacks	4.63 ± 0.00AC	4.63–4.63	0.41 ± 0.00A	0.41–0.41	137.38 ± 0.00A	137.38–137.38
Fats and oils	1.50 ± 0.50A	0.02–3.18	0.60 ± 0.07AB	0.45–0.78	12.76 ± 6.14A	5.20–37.28
Honey and sugar	0.87 ± 0.48A	0.22–1.80	0.57 ± 0.13AB	0.40–0.82	5.63 ± 3.95A	1.63–13.53
Condiments	2.38 ± 0.66AC	1.02–5.08	1.19 ± 0.41AB	0.11–3.11	74.61 ± 33.78A	23.48–242.18
Drinking water	4.58 ± 0.00AC	4.58–4.58	1.06 ± 0.00AB	1.06–1.06	39.83 ± 0.00A	39.83–39.83
**Food Group**	**Elements**					
	**Ca**		**P**		**Na**	
	**Mean ± SE**	**Min–Max**	**Mean ± SE**	**Min–Max**	**Mean ± SE**	**Min–Max**
Bread	127.10 ± 19.81AC	73.79–166.24	391.46 ± 91.93AB	173.39–591.18	2079.90 ± 370.62A	968.08–2458.31
Cereals and cereal products	54.08 ± 13.61A	11.34–81.68	132.79 ± 10.86AE	97.03–156.58	1397.17 ± 316.31A	176.21–2005.87
Legumes	233.14 ± 44.43AC	150.69–376.67	598.15 ± 117.04BD	329.75–948.48	562.21 ± 428.55A	57.73–2272.56
Red meat and meat products	171.55 ± 31.83AC	62.74–323.12	946.92 ± 86.12C	643.83–1298.31	2349.82 ± 264.48A	1487.21–3703.18
Poultry	195.06 ± 99.03AC	96.03–294.10	1126.40 ± 55.56C	1070.85–1181.96	2142.96 ± 259.29A	1883.67–2402.25
Fish	190.14 ± 113.32AC	76.82–303.461	828.18 ± 300.96CD	527.22–1129.14	2329.85 ± 308.10A	2021.75–2637.96
Eggs	253.88 ± 0.00ABC	253.88–253.88	328.01 ± 0.00ABE	328.01–328.01	658.98 ± 0.00A	658.98–658.98
Milk and milk products	543.43 ± 107.67B	215.89–1520.68	283.11 ± 45.93AE	131.53–618.78	1106.35 ± 388.36A	138.45–4081.93
Raw vegetables	214.32 ± 51.65AC	66.29–542.69	98.59 ± 19.43AE	41.88–226.18	306.70 ± 139.53A	14.94–1232.51
Cooked vegetables	250.29 ± 63.06ABC	55.24–532.66	130.52 ± 27.83AE	45.80–229.63	434.63 ± 189.77A	3.09–1806.61
Potatoes	60.94 ± 0.00A	60.94–60.94	177.20 ± 77.43AE	99.78–254.63	751.54 ± 633.03A	118.51–1384.56
Fruits	89.23 ± 11.74AC	28.94–199.01	53.23 ± 9.88AE	7.23–239.48	200.39 ± 16.79A	3.76–273.00
Fruit juices and soft beverages	208.14 ± 133.60AC	6.09–737.34	41.71 ± 19.10AE	9.03–108.76	221.42 ± 60.28A	142.21–459.93
Nuts and dried fruits	405.68 ± 227.03BC	7.26–1292.95	813.41 ± 361.20CD	254.95–2187.86	589.81 ± 289.69A	123.16–1642.49
Cookies, cakes, and pastry	128.46 ± 25.66AC	83.99–246.44	288.75 ± 47.81ABE	174.35–494.00	1035.02 ± 256.15A	357.26–2178.56
Snacks	218.64 ± 0.00AC	218.64–218.64	341.48 ± 0.00ABE	341.48–341.48	2444.66 ± 0.00A	2444.66–2444.66
Fats and oils	69.42 ± 28.24AC	25.94–178.59	71.03 ± 42.0AE	1.43–235.18	178.90 ± 52.43A	7.05–308.31
Honey and sugar	24.38 ± 1.15A	22.14–25.94	12.36 ± 5.63E	4.38–23.23	165.48 ± 17.76A	131.71–191.90
Condiments	103.89 ± 18.96AC	48.54–165.99	79.80 ± 45.56AE	22.65–307.08	37466.54 ± 35475.77B	227.25–214836.25
Drinking water	203.19 ± 0.00AC	203.19–203.19	27.70 ± 0.00AE	27.70–27.70	264.30 ± 0.00A	264.30–264.30

Different capital letters in the same column indicate significant differences (*P* < 0.05) between food groups.

**Table 4 nutrients-12-03370-t004:** The mean, minimum, and maximum daily intakes (mg/d) of zinc (Zn), copper (Cu), magnesium (Mg), calcium (Ca), phosphorus (P), and sodium (Na) from different food groups in adult population in Shiraz, Iran.

Food Group	Elements					
	Zn		Cu		Mg	
	Mean ± SD	Min–Max	Mean ± SD	Min–Max	Mean ± SD	Min–Max
Bread	1.27 ± 0.82A	0.07–9.08	0.21 ± 0.14A	0.01–1.73	31.47 ± 20.21A	1.83–216.75
Cereals and cereal products	1.09 ± 0.55B	0.03–3.09	0.30 ± 0.15B	0.01–1.17	19.26 ± 9.65B	0.56–65.03
Legumes	0.34 ± 0.28CF	0.02–2.71	0.09 ± 0.07CHK	0–0.73	19.38 ± 17.04B	0.70–184.05
Red meat and meat products	1.80 ± 1.65D	0.02–12.80	0.15 ± 0.34D	0.00–6.55	15.32 ± 14.17C	0.12–135.28
Poultry	0.25 ± 0.20CK	0.00–1.78	0.03 ± 0.03EI	0.00–0.24	8.15 ± 6.87D	0.00–63.16
Fish	0.06 ± 0.06EJ	0.00–0.47	0.01 ± 0.01E	0.00–0.09	3.03 ± 3.46EJ	0.00–27.76
Eggs	0.45 ± 0.71F	0.00–6.80	0.02 ± 0.03E	0.00–0.26	11.77 ± 18.74F	0.00–178.21
Milk and milk products	1.04 ± 0.80B	0.06–7.67	0.34 ± 0.26F	0.02–2.38	61.50 ± 50.41G	1.79–435.90
Raw vegetables	0.58 ± 0.39G	0.05–3.29	0.38 ± 0.25G	0.01–2.09	80.95 ± 55.05H	1.26–490.68
Cooked vegetables	0.18 ± 0.12HK	0.01–0.87	0.12 ± 0.07H	0.01–0.58	25.03 ± 15.82I	1.37–128.62
Potatoes	0.10 ± 0.17EHJ	0.00–3.09	0.06 ± 0.11CI	0.00–1.92	5.33 ± 10.98DJK	0.00–200.29
Fruits	0.79 ± 0.47I	0.00–3.57	0.80 ± 0.46J	0.00–3.30	61.21 ± 40.18G	0.00–317.42
Fruit juices and soft beverages	0.04 ± 0.05E	0.00–0.42	0.02 ± 0.03E	0.00–0.31	2.60 ± 5.47EJ	0.00–52.22
Nuts and dried fruits	0.19 ± 0.32HK	0.00–4.57	0.09 ± 0.12HK	0.00–1.37	17.29 ± 36.68BC	0.00–536.96
Cookies, cakes, and pastry	0.17 ± 0.20HKJ	0.00–1.87	0.06 ± 0.08CIK	0.00–0.65	4.66 ± 5.97JK	0.00–59.89
Snacks	0.02 ± 0.05E	0.00–0.93	0.00 ± 0.00E	0.00–0.08	0.45 ± 1.63E	0.00–27.48
Fats and oils	0.03 ± 0.04E	0.00–0.70	0.01 ± 0.01E	0.00–0.21	0.22 ± 0.27E	0.00–2.99
Honey and sugar	0.01 ± 0.01E	0.00–0.05	0.01 ± 0.01E	0.00–0.05	0.08 ± 0.10E	0.00–0.88
Condiments	0.18 ± 0.12HK	0.00–1.01	0.10 ± 0.07H	0.00–0.60	7.20 ± 5.16DK	0.00–41.74
Drinking water	4.34 ± 2.92L	0.00–31.57	1.01 ± 0.68L	0.00–7.33	37.76 ± 25.43L	0.00–274.79
Total diet	12.92 ± 4.70	4.65–42.67	3.80 ± 1.37	1.19–11.52	412.68 ± 147.80	165.61–1045.65
**Food Group**	**Elements**					
	**Ca**		**P**		**Na**	
	**Mean ± SD**	**Min–Max**	**Mean ± SD**	**Min–Max**	**Mean ± SD**	Min–Max
Bread	15.54 ± 9.76AF	0.93–91.78	64.23 ± 40.99A	3.44–386.70	255.85 ± 199.55A	10.87–2819.86
Cereals and cereal products	10.62 ± 5.43AC	0.40–44.88	38.35 ± 19.19B	1.04–122.75	396.07 ± 200.32B	10.73–1259.37
Legumes	8.09 ± 6.43BC	0.34–56.21	16.95 ± 13.74C	0.61–128.19	14.81 ± 17.99CDEF	0.11–212.93
Red meat and meat products	7.10 ± 7.04BCG	0.07–83.44	43.50 ± 41.26D	0.71–516.91	96.26 ± 84.65G	2.22–920.05
Poultry	3.20 ± 3.09BGIJ	0.00–36.32	35.84 ± 28.30B	0.00–207.20	58.10 ± 47.00H	0.00–367.18
Fish	2.74 ± 3.22BGIJ	0.00–26.11	10.68 ± 12.21EF	0.00–97.82	26.33 ± 29.52F	0.00–230.49
Eggs	10.06 ± 16.02C	0.00–152.33	13.00 ± 20.69E	0.00–196.80	26.11 ± 41.57F	0.00–395.39
Milk and milk products	136.44 ± 92.04D	6.16–855.45	62.17 ± 42.70A	5.39–344.34	240.74 ± 199.79I	5.82–2387.98
Raw vegetables	58.01 ± 39.76E	1.08–339.60	19.59 ± 13.33C	0.87–117.39	25.17 ± 22.47F	0.76–262.71
Cooked vegetables	20.19 ± 13.19F	1.06–90.14	8.58 ± 5.44F	0.69–38.02	21.14 ± 16.50EF	0.38–140.41
Potatoes	1.76 ± 3.17GIJ	0.00–56.00	4.54 ± 11.94G	0.00–224.04	17.01 ± 63.04DEF	0.00–1191.08
Fruits	47.55 ± 28.34H	0.00–202.92	28.12 ± 16.70H	0.00–106.25	118.96 ± 68.79J	0.00–488.47
Fruit juices and soft beverages	3.82 ± 6.59BGIJ	0.00–57.32	1.26 ± 2.08I	0.00–19.53	6.15 ± 8.80CD	0.00–77.15
Nuts and dried fruits	1.92 ± 3.71GIJ	0.00–57.99	10.43 ± 18.60EF	0.00–261.52	7.53 ± 14.53CDE	0.00–196.17
Cookies, cakes, and pastry	3.61 ± 4.53BGIJ	0.00–39.43	8.89 ± 11.37F	0.00–112.16	28.79 ± 35.20F	0.00–343.59
Snacks	0.72 ± 2.60I	0.00–43.73	1.12 ± 4.06I	0.00–68.30	7.99 ± 29.05CDE	0.00–488.93
Fats and oils	1.20 ± 1.36IJ	0.00–14.03	1.01 ± 1.46I	0.00–10.72	4.47 ± 5.18CD	0.00–80.78
Honey and sugar	0.34 ± 0.29I	0.00–2.58	0.17 ± 0.18I	0.00–1.62	2.24 ± 1.92C	0.00–16.70
Condiments	7.01 ± 4.72BCGJ	0.00–39.66	8.10 ± 6.06F	0.00–47.16	1239.03 ± 734.95J	0.00–6875.20
Drinking water	192.68 ± 129.76K	0.00–1401.99	26.27 ± 17.69H	0.00–191.13	250.63 ± 168.78AI	0.00–1823.67
Total diet	532.61 ± 204.55	125.41–1832.56	402.77 ± 135.78	180.03–1093.15	2843.39 ± 1026.13	803.88–9743.65

Different capital letters in the same column indicate significant differences (*P* < 0.05) between food groups.

**Table 5 nutrients-12-03370-t005:** Estimated dietary intakes of zinc (Zn), copper (Cu), magnesium (Mg), calcium (Ca), phosphorus (P), and sodium (Na) in adult population (*n* = 438, 199 males and 239 females) in Shiraz, Iran and comparison with guidance values.

Elements	Dietary Exposure (mg/d)				Mean Contribution to		PCT2.5 Contribution to EAR (%)	PCT97.5 Contribution to UL (%)	<EAR% ^a^	>UL% ^b^
	PCT2.5	PCT50	PCT97.5	Mean ± SD	EAR (%)	UL (%)				
Zn	Male:7.61	14.54	23.91	15.12 ± 5.17	160.8	37.8	81.0	59.8	9.1	0.2
	Female:5.51	10.86		11.10 ± 3.32	163.2	27.8	81.0			
Cu	1.70	3.54	7.28	3.80 ± 1.37	542.8	38.0	242.8	72.8	0	0.5
Mg	Male: 211.88	416.64	792.03	455.68 ± 165.10	130.2	NA	60.5	NA	22.4	NA
	Female: 196.99	354.28		376.87 ± 120.84	142.2		74.3			
Ca	244.67	494.50	1058.92	532.61 ± 204.55	66.6	21.3	30.6	42.4	91.6	0
P	205.38	379.01	709.59	402.77 ± 135.78	69.4	10.1	35.4	17.7	89.7	0
Na *	1469.10	2646.07	5470.40	2843.39 ± 1026.13	189.6	123.6	97.9	237.8	0.5	69.6

^a^ Percentage of population with intakes lower than EAR. ^b^ Percentage of population with intakes above the UL. * Adequate Intake was used instead of EAR (lack of sufficient scientific evidence to establish an EAR). NA, not applicable as no UL is set for dietary Mg intake; PCT, Percentile; EAR, Estimated Average Requirement; UL, Tolerable Upper Intake Level.

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
