# Peer review of "Dietary Intakes of Zinc, Copper, Magnesium, Calcium, Phosphorus, and Sodium by the General Adult Population Aged 20–50 Years in Shiraz, Iran: A Total Diet Study Approach"

_nutrients, 2020, doi:10.3390/nu12113370_

Round 1
Reviewer 1 Report
The manuscript “Dietary intakes of zinc, copper, magnesium, calcium, phosphorous, and sodium by the general population in Shiraz, Iran: A total diet study approach ” fits well with the aims and scope of the journal. The study is well organized and proposes original and practical results. However, some drawbacks related to the introduction part are present. For this reason, I recommend a major revision. Comments follow.
The introduction part lacks a broad overview on the importance of detection of inorganic elements in food by ICP techniques (not only OES but also MS), and their consequent relevance on diet.
To this purpose you may cite recent works such as
- Cicero, N., Albergamo, A., Salvo, A., Bua, G. D., Bartolomeo, G., Mangano, V., ... & Dugo, G. (2018). Chemical characterization of a variety of cold-pressed gourmet oils available on the Brazilian market. Food Research International, 109, 517-525.
-Albergamo, A., Bua, G. D., Rotondo, A., Bartolomeo, G., Annuario, G., Costa, R., & Dugo, G. (2018). Transfer of major and trace elements along the “farm-to-fork” chain of different whole grain products. Journal of Food Composition and Analysis, 66, 212-220.
-Bella, G. D., Licata, P., Potortì, A. G., Crupi, R., Nava, V., Qada, B., ... & Turco, V. L. (2020). Mineral content and physico-chemical parameters of honey from North regions of Algeria. Natural Product Research, 1-8.
Reviewer 2 Report
Thank you very much for allowing me to review the article "Dietary intakes of zinc, copper, magnesium, calcium, phosphorous, and sodium by the general population in Shiraz, Iran: A total diet study approach" (nutrients-972417).
It is based on the fact that the diet is a source of nutrients and contaminants and its crucial contributor to human health and disease. Also its base on the lack data on the nutrient content of Iranian food.
They study the total diet, with special attention on the dietary intake of zinc (Zn), copper (Cu), magnesium 22 (Mg), calcium (Ca), phosphorous (P), and sodium (Na) by healthy adults in Shiraz , Iran. It was estimated from the foods as consumed on 580 individual food items were collected, prepared, and pooled into 129 composite samples.
Comments:
Title, it should be adjusted to the population they really study, which is about individuals between 20 and 50 years old, this is not the total population
1.-Introduction: The work is well justified but it lacks information on the nutritional elements that are studied. It should be completed with updated information on these nutrients, so that the reader can better understand the work.
2.-Materials and methods
They used the consumption data on 438 healthy adults (199 males and 239 females) aged 20-50 years old using a 160-item FFQ.
Please explain if this sample is representative of the Iranian population, it is proportional for men and women to the population distribution.
Has a sample size been calculated for this study?
Table 2 corresponds to material and methods but the rest are results.
3.-Results: They should have their own section.
Tables and Figures are very informative, but the tables should be formatted so that they can be better evaluated, more compressed, also in the base it should be indicated which foods in each category have been studied, for example fish, indicate which fish has been analyzed.
4.-Discussion: It is well raised but the limitations of the study should be further distinguished.
Conclusion: It is concluded that daily salt intake in the Iranian population substantially exceeds the current recommendations. Also to Zn, Cu, and Mg, the prevalence of inadequate or excessive intakes was too low to be of major concern. Meanwhile, a very high proportion of the subjects (more than 75%) were at risk of inadequate intake of Ca or P mainly due to low consumption of animal foods including milk and dairy products.
Altogether it is a very interesting study that provides information that will guide future nutritional improvement measures and serve as a basis for future studies.
Round 2
Reviewer 1 Report
The suggestions were accepted. The manuscript can be accepted in the present version